# Ultrasound-Guided Needle Aspiration Biopsy of Superficial Metastasis of Lung Cancer with and without Rapid On-Site Evaluation: A Randomized Trial

**DOI:** 10.3390/cancers14205156

**Published:** 2022-10-21

**Authors:** Vanina Livi, Giovanni Sotgiu, Alessandra Cancellieri, Daniela Paioli, Fausto Leoncini, Daniele Magnini, Rocco Trisolini

**Affiliations:** 1Interventional Pulmonology Unit, Fondazione Policlinico Universitario A. Gemelli IRCCS, 00168 Rome, Italy; 2Clinical Epidemiology and Medical Statistics Unit, Department of Medicine, Surgery and Pharmacy, University of Sassari, 07100 Sassari, Italy; 3Pathology Unit, Fondazione Policlinico Universitario A. Gemelli IRCCS, 00168 Rome, Italy; 4Department of Cardiovascular and Pulmonary Sciences, Catholic University of the Sacred Hearth, 00168 Rome, Italy

**Keywords:** lung cancer, bronchoscopy and interventional techniques, pathology, radiology and other imaging

## Abstract

**Simple Summary:**

Pulmonologist-performed US-NAB of “superficial” metastatic lesions is safe and has an excellent diagnostic yield for both tissue diagnosis and molecular profiling regardless of the use of rapid on-site evaluation. These findings have important implications for costs, hospital resource allocation, and the globally widespread utilization of US-NAB.

**Abstract:**

Background and Objective: Studies which evaluated the role of an ultrasound-guided needle aspiration biopsy (US-NAB) of metastases from lung cancer located in “superficial” organs/tissues are scant, and none of them assessed the possible impact of rapid on-site evaluation (ROSE) on diagnostic accuracy and safety outcomes. Methods: Consecutive patients with suspected superficial metastases from lung cancer were randomized 1:1 to US-NAB without (US-NAB group) or with ROSE (ROSE group). The diagnostic yield for a tissue diagnosis was the primary outcome. Secondary outcomes included the diagnostic yield for cancer genotyping, the diagnostic yield for PD-L1 testing, and safety. Results: During the study period, 136 patients were randomized to receive an US-NAB with (*n* = 68) or without ROSE (*n* = 68). We found no significant differences between the ROSE group and the US-NAB group in terms of the diagnostic yields for tissue diagnosis (94.1% vs. 97%, respectively; *p* = 0.68), cancer genotyping (88% vs. 91.8%, respectively; *p* = 0.56), and PD-L1 testing (93.5% vs. 90.6%, respectively; *p* = 0.60). Compared to the diagnostic US-NAB procedures, the non-diagnostic procedures were characterized by less common use of a cutting needle (66.6% vs. 96.9%, respectively; *p* = 0.0004) and less common retrieval of a tissue core (37.5% vs. 98.5%; *p* = 0.0001). Only one adverse event (vasovagal syncope) was recorded. Conclusion: US-NAB of superficial metastases is safe and has an excellent diagnostic success regardless of the availability of ROSE. These findings provide a strong rationale for using US-NAB as the first-step method for tissue acquisition whenever a suspected superficial metastatic lesion is identified in patients with suspected lung cancer.

## 1. Introduction

Personalized oncological treatments have markedly improved the response rate, quality of life, and survival of patients with locally advanced and advanced non-small-cell lung cancer [1,2,3]. However, despite the availability of highly effective endoscopy-based and CT-based biopsy procedures, 3% to 52% of patients who are potentially eligible for targeted therapies or immunotherapies do not have access to a diagnosis or to thorough and/or timely molecular profiling for a variety of reasons, including poor performance status, unfavorable tumor location, and poor quality of the biopsy samples [4,5,6,7,8].

Lung cancer metastases located in superficial organs/tissues are ideal targets for an ultrasound-guided needle aspiration biopsy (US-NAB), which is less invasive and costly than both advanced diagnostic bronchoscopy and CT-guided biopsy. In the specific setting of superficial lesions, the role of US-NAB has been assessed in a limited proportion of the patient population in a handful of observational studies [9,10,11]. Interestingly, the diagnostic yield (54% to 86%) reported in studies in which US-NAB was performed without rapid on-site evaluation (ROSE) [9,10] was lower than that obtained in the only study in which ROSE was systematically used (98%) [11]. While the above results may point to the possible added value of ROSE, data obtained in other clinical settings suggest caution. In particular, several randomized controlled trials assessing the value of ROSE performed on specimens retrieved endoscopically from lymphadenopathy [12,13,14,15] have failed to confirm the increase in diagnostic yield associated with its use that had previously been suggested by observational studies [16,17,18].

The aim of the present randomized study is to assess the impact of ROSE on diagnostic accuracy and potential complications in consecutive patients with suspected lung cancer showing a possible superficial metastatic lesion on imaging studies (CT and/or PET).

## 2. Methods

### 2.1. Study Design, Setting, and Participants

This randomized controlled trial was approved by the Ethics Committee of the Fondazione Policlinico Universitario Agostino Gemelli IRCCS (Prot. ID 3866) and was registered at ClinicalTrial.gov (NCT04618874). 

Individuals were eligible if they were >18 years old, had a suspected superficial metastasis from lung cancer on CT and/or PET, and had an indication for tissue acquisition. The key exclusion criteria were as follows: unwillingness or inability to consent; evidence of a non-correctable coagulation disorder; and use of antiplatelet (excluding aspirin) or anticoagulant drugs that could not be discontinued.

### 2.2. Randomization and Interventions

Consecutive patients were randomized 1:1 to US-NAB without (US-NAB group, standard care) or with ROSE (ROSE group, intervention). The randomization sequence was computer-generated and we placed the assignments in opaque, sealed envelopes. Owing to the nature of the interventions, the participants and the investigators could not be blinded to the group allocation. The recruitment took place between 1 April 2021 and 31 March 2022.

All study procedures were carried out by a team of 5 interventional pulmonologists. Each patient underwent an ultrasound examination using an Hitachi Aloka Arietta 850 ultrasound platform (Steinhausen, Switzerland), and the “target” lesion was evaluated with both a convex and a linear probe. In cases in which sampling was considered feasible and safe (i.e., lack of interposition of large vessels), the skin was sterilized and 2–5 mL of 2% lidocaine was injected into the subcutaneous area of interest. Then, the lesion was sampled using a “freehand” technique with a 22-gauge needle Farmatexa Softouch syringe (Farmac-Zabban, Calderara Di Reno, Bologna, Italy). 

In the US-NAB arm, the specimen obtained with the first needle pass was smeared on 2–4 glass slides, which were placed into a 96% alcohol solution and were later stained with the Papanicolaou method in the Pathology lab. Then, 2 to 4 more needle passes were performed, at the discretion of the operator, depending on the size of the lesion, the needle type used, and the gross appearance of the material retrieved. After the first needle pass, an 18G or 16G cutting needle (Biomol biopsy set, Hospital Service SPA, Rome, Italy) was used, whenever considered possible and safe.

In the ROSE arm, the specimen obtained with the first needle pass was smeared on 2–4 glass slides. One slide was air-dried, stained with Diff-Quik, and submitted to ROSE, whereas the remaining were put into an ethanol solution and sent to the Pathology lab. Further sampling was guided by the ROSE results. If successful sampling (i.e., representativeness of the lesion’s tissue or clues to a specific pathologic diagnosis) was confirmed, 2 to 4 more needle passes were performed, depending on the size of the lesion and the gross appearance of the material retrieved, using a cutting needle whenever considered possible and safe. 

In patients with more than one superficial lesion, the decision to sample one or more lesions was left at the discretion of the operator, who could decide based on the ROSE findings (ROSE arm) or on the gross examination of the biopsy material (US-NAB arm). 

After the procedure, each patient was monitored in the recovery room for at least 1 h before being discharged (outpatients) or sent back to their hospital ward (inpatients).

Molecular profiling included testing for PD-L1 and for the genes reimbursed at the time of biopsy by the Italian Healthcare System (i.e., *EGFR*, *ALK*, *ROS1*, *KRAS*, and *BRAF*). A minimum of 30 ng/L of DNA and 15 ng/L of RNA were deemed sufficient to carry out the above analysis.

### 2.3. Outcomes

The primary outcome was the diagnostic yield of the US-NAB. Secondary outcomes were the following: (a) the diagnostic yield for cancer genotyping (successful completion of *EGFR*, *KRAS*, *ALK*, *ROS1*, and *BRAF* testing); (b) the diagnostic yield for PD-L1 testing; and (c) safety (percentage of adverse events). 

### 2.4. Sample Size Calculation and Statistical Analysis

Based on the limited data from the observational studies available in the literature [9,10,11], the diagnostic yield of US-NAB with ROSE [11] was expected to be higher than that of US-NAB without ROSE [9,10] by 20% (90% vs. 70%). Considering a type I error α of 5%, a power 1 − β of 80%, and a drop-out rate of 10%, a sample size of 136 patients was estimated.

Qualitative variables were described with absolute and relative (percentage) frequencies, whereas quantitative variables were described with means (standard deviations) or medians (interquartile ranges), depending on their normal or non-normal distribution. Statistical differences between qualitative variables were evaluated using chi-squared or Fisher’s exact test, as appropriate. Student’s *t*-test and Mann–Whitney test were performed to assess statistically significant differences for normally and non-normally distributed quantitative variables, respectively. A two-tailed *p*-value less than 0.05 was considered statistically significant. Data were analyzed using STATA, version 16 (StatsCorp, College Station, TX, USA). 

## 3. Results

Between 1 April 2021 and 31 March 2022, 144 patients were screened and 136 were enrolled in the study (Figure 1). Table 1, which displays the baseline characteristics of the patients and the procedures, shows no significant differences in demographics and clinical characteristics. One hundred forty-four lesions belonging to several superficial anatomical areas were sampled (Figure 2 and Figure 3), but superficial lymph nodes (103, 71.5%) and bone metastases (18, 12.5%) were the most frequent targets. The final diagnosis was metastasis from a pulmonary tumour in the majority of patients (121, 89%), with adenocarcinoma (90, 66.2%) and small-cell lung cancer (15, 11%) being the most prevalent histologic subtypes.

Table 2 shows the results of the study endpoints. We found no significant differences between the ROSE group and the US-NAB group in terms of the diagnostic yield for tissue diagnosis (94.1% vs. 97%, respectively; *p* = 0.68). Subgroup analyses did not show any significant differences in the diagnostic yield according to the location of the target lesion or to the final histologic diagnosis. 

The diagnostic yields for both cancer genotyping and PDL1 testing, which were requested by the treating oncologist in 91 and 99 patients, respectively, were excellent in both study groups, with no statistically significant differences (Table 2). 

Table 3 shows the main demographic, clinical, and procedural characteristics of the six patients in whom the US-NAB procedure was non-diagnostic. Comparing patients with a diagnostic and a non-diagnostic US-NAB procedure, we found a statistically significant difference in regard to the following two technical characteristics. A cutting needle (16G or 18G) was used more frequently in patients with a diagnostic procedure than in those with a non-diagnostic US-NAB procedure (126/130, 96.9% vs. 4/6, 66.6%; *p* = 0.0004). A tissue core was obtained more frequently from the 136 lesions sampled in the 130 patients with a diagnostic US-NAB than from the eight lesions sampled in the six patients with a non-diagnostic procedure (134/136, 98.5% vs. 3/8, 37.5%; *p* = 0.0001).

A single adverse event (vasovagal syncope) was observed during the study.

## 4. Discussion

The present study found that US-NAB of superficial metastases from lung cancer has high diagnostic success and safety regardless of the availability of ROSE. These findings, which confirm the importance using randomized trials to verify the hypothesis raised by observational studies, have important potential implications for costs, allocation of healthcare resources, and the widespread utilization of US-NAB sampling for this specific indication.

The diagnostic yields for tissue diagnosis and molecular profiling that we achieved are at least similar to those reported for advanced endoscopy-based and CT-based biopsy methods [19,20,21,22,23] but with a much better safety profile [23,24,25,26]. No currently available studies aiming at assessing the role of US-NAB for the diagnosis of superficial metastases reported clinically important complications [9,10,11]. This is not surprising, as US-NAB is performed following a sterile protocol, under real-time ultrasound control, and after an evaluation of the availability of a safe, vessel-free path to the lesion. Furthermore, sampling a superficial metastatic target significantly reduces the risk of a pneumothorax.

Besides its high success rate and safety regardless of the use of on-site cytologic review, US-NAB of a superficial lesion is associated with several other potential organizational, economic, and ethical advantages. It does not require sedation, does not expose patients and health care staff to radiation, can be performed safely in patients with poor performance status, and is less invasive and costly than endoscopic or CT-guided procedures [27,28,29]. Furthermore, respiratory physicians, who are frequently involved in the diagnostic and staging pathway of lung cancer patients, may rapidly schedule and perform an US-NAB when indicated, and may decide for more invasive procedures when the US-NAB is inconclusive [30]. However, there is a relative lack of knowledge of these potential advantages among the stakeholders involved in the lung cancer pathway, and US-NAB is not even mentioned among the diagnostic methods for diagnosing or staging NSCLC by most international oncological and thoracic scientific societies [31,32,33]. This is confirmed by the fact that most of the patients that we enrolled in the present study were referred to our division for bronchoscopy and underwent an US-NAB instead after we carefully reviewed their imaging studies and identified a suspected superficial metastasis.

Besides lymph nodes, bone lesions were the most common targets in our study. Remarkably, we observed the presence of metastatic lesions completely replacing the bone structure in a non-negligible percentage of patients with advanced lung cancer. In most such cases, the needle found no resistance in penetrating the lesion and the histological examination of the biopsy specimen showed almost no residual bone tissue. In the few cases in which we had to biopsy a metastatic lesion within a partially intact bone, we agreed with the pathology department that the specimen did not undergo decalcification. The latter, in fact, is known to cause the denaturation of the nucleic acids and may compromise the accomplishment of a reliable cancer genotyping [34]. 

Although the small number of non-diagnostic procedures weakens the strength of the comparison with diagnostic US-NABs, the use of a cutting needle (16G or 18G) and the retrieval of tissue cores were significantly more common in the diagnostic cases. This is not surprising as tissue cores allow the pathologist to assess the tissue architecture and can make it easier to diagnose with confidence those conditions which may only be suspected on cytological grounds.

The randomized design and the variety of superficial lesions that were sampled represent the strengths of our study. However, some limitations that possibly affect the study’s external validity should be acknowledged. ROSE was performed by a pathologist (A.C.) with extensive experience in pulmonary pathology and on-site cytologic review in both clinical and research settings [12,35,36,37]. Furthermore, more than 200 US-NABs from superficial lesions are carried out annually in our center by experienced operators. The higher percentage of patients in whom a cutting needle was used in this study, in comparison to that of an observational study [11] we performed only a few years ago in the same clinical setting (95.6% vs. 71%, respectively), suggests a higher level of confidence of the operators in the management of superficial lesions of any size. 

## 5. Conclusions

In conclusion, our results highlight the importance of carefully reviewing the imaging of patients with suspected lung cancer who undergo evaluation for a biopsy, and the importance of using US-NAB as the first-step method whenever a suspected superficial metastatic lesion is identified.

## Figures and Tables

**Figure 1 cancers-14-05156-f001:**
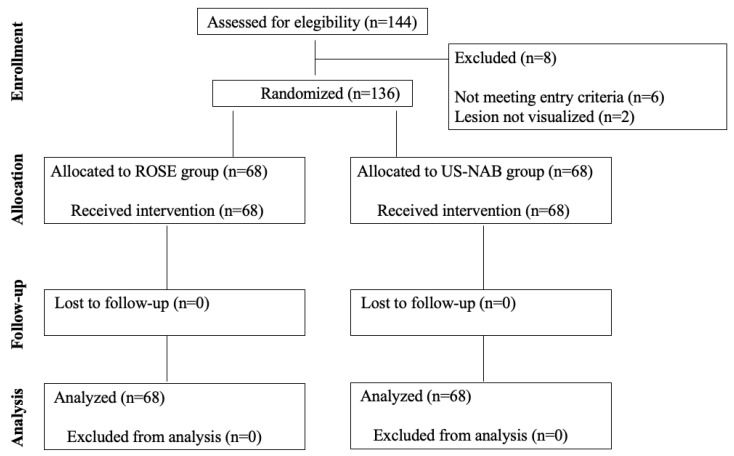
Consort flow diagram of the study.

**Figure 2 cancers-14-05156-f002:**
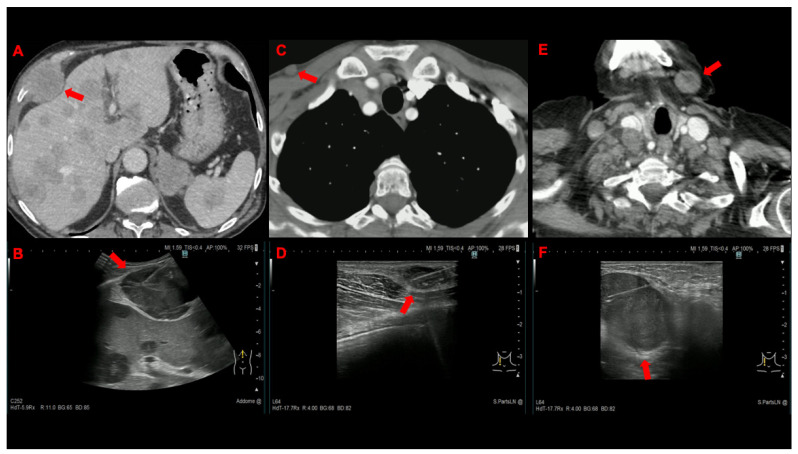
Contrast-enhanced CT (**A**) and corresponding US-NAB image (**B**) showing a large thoracic wall lesion (red arrow) adjacent to the liver, which contains multiple nodular lesions consistent with metastases; contrast-enhanced CT (**C**) and corresponding US-NAB image (**D**) showing a sub-centimetric subcutaneous nodule (red arrow) in the anterior aspect of the right hemithorax; and contrast-enhanced CT (**E**) and corresponding US-NAB image (**F**) from an enlarged, round cervical lymph node (red arrow).

**Figure 3 cancers-14-05156-f003:**
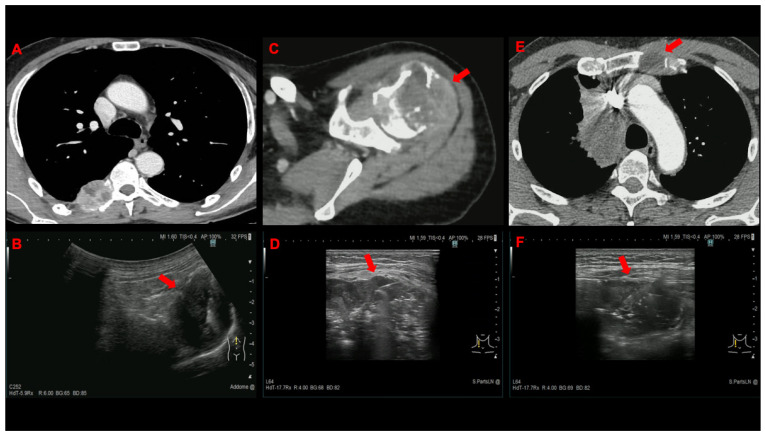
Contrast-enhanced CT and corresponding US-NAB image of US-NAB (red arrow) from bone metastasis involving a rib and a transvers process of a thoracic vertebra (**A**,**B**), the right homer (**C**,**D**), and the sternum (**E**,**F**).

**Table 1 cancers-14-05156-t001:** Demographics and baseline characteristics of the patients (136) and lesions (144).

	ROSE Group	US-NAB Group
**Mean (SD) age, years**	67.6 (10.3)	67.3 (8.5)
**Gender, *n*. (%)**		
Female	30 (44)	28 (41)
Male	38 (56)	40 (59)
**Smoking history, *n*. (%)**		
Current	19 (30)	21 (31)
Former	35 (51.5)	36 (53)
Never	14 (20.5)	11 (16)
**Lesions sampled, *n*.**	72	72
**Median (IQR) lesion size, mm**		
Short axis	13.2 (9.2–18.3)	12.4 (9–16.1)
Long axis	18.2 (12.4–25.1)	16.5 (12.8–26.6)
**Metastatic site, *n*. (%)**		
** *Lymph node* **	54 (75)	49 (68)
*Supraclavicular*	44	40
*Cervical*	8	7
*Axillary*	2	1
*Mammary*	0	1
** *Bone* **	7 (9.7)	11 (15.3)
** *Subcutaneous tissue* **	6 (8.3)	6 (8.3)
** *Thoracic wall* **	3 (4.2)	3 (4.2)
** *Muscle* **	2 (2.8)	1 (1.4)
** *Pleura* **	0 (0)	2 (2.8)
**Needle gauge, *n*. (%)**		
22G	4 (5.9)	2 (3)
22G + 18G	53 (77.9)	60 (88.2)
22G + 16G	11 (16.2)	6 (8.8)
**Tissue core retrieval, *n*. (%)**	64 (94.1)	66 (97)
**Final diagnosis per patient, *n*. (%)**		
Adenocarcinoma	41 (60.3)	49 (72)
Squamous cell carcinoma	4 (5.9)	1 (1.5)
Small-cell lung cancer	11 (16)	4 (5.9)
Large cell neuroendocrine carcinoma	1 (1.5)	0 (0)
Carcinoid tumor	1 (1.5)	1 (1.5)
NSCLC NOS	4 (5.9)	4 (5.9)
Metastasis from extrapulmonary tumor	3 (4.5)	5 (7.4)
Lymphoprolipherative disorder	2 (2.9)	2 (2.9)
Tuberculosis	0 (0)	2 (2.9)
Reactive lymphadenopathy	1 (1.5)	0 (0)

**Table 2 cancers-14-05156-t002:** Study outcomes.

Outcomes	ROSE Group (No. 68)	US-NAB Group (No. 68)	*p*-Value
**Primary outcome**			
Diagnostic yield for tissue diagnosis	64/68 (94.1%)	66/68 (97%)	0.68
**Secondary outcome**			
Diagnostic yield for cancer genotyping ^	37/42 (88%)	45/49 (91.8%)	0.56
Diagnostic yield for PD-L1 testing ^	43/46 (93.5%)	48/53 (90.6%)	0.60
Complications	1/68 (1.5%)	0/68 (0.0%)	1.00

^ Molecular profiling results are given for patients in whom genotyping and/or PDL1 testing were requested by the medical oncologist (see Section 3).

**Table 3 cancers-14-05156-t003:** Demographic, clinical, and procedural characteristics in the six patients with a non-diagnostic US-NAB.

	Study Group	Age/Sex	Lesion Location	No. Lesions Sampled	Needle Used	Tissue Core Retrieved	Final Diagnosis
#1	ROSE	77/M	LN	1	18G	No	NSCLC NOS
#2	ROSE	73/M	LN	1	22G	No	SQCC
#3	ROSE	66/F	LN	2	18G	Yes	Adenocarcinoma
#4	ROSE	48/M	LN	2	22G	No	Adenocarcinoma
#5	US-NAB	66/F	LN	1	18G	Yes	Adenocarcinoma
#6	US-NAB	86/M	Bone	1	18G	Yes	Adenocarcinoma

Abbreviations: LN = lymph node; NSCLC NOS = non-small cell lung cancer not otherwise specifiable; SQCC = squamous cell carcinoma.

## Data Availability

The data can be shared up on request.

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
