# Peer review of "Ultrasound-Guided Needle Aspiration Biopsy of Superficial Metastasis of Lung Cancer with and without Rapid On-Site Evaluation: A Randomized Trial"

_cancers, 2022, doi:10.3390/cancers14205156_

Round 1
Reviewer 1 Report
This is a clearly written and well carried out study that addresses an important issue in daily management of patients and strongly argues that costly ROSE may not be necessary in this patient group.
Informations on what molecular analysis was carried out and how much DNA or RNA was needed ist important to understand whether Needle Aspiration Biopsy yielded enough tissue for a full work up.
Author Response
Comment 1: Informations on what molecular analysis was carried out and how much DNA or RNA was needed its important to understand whether Needle Aspiration Biopsy yielded enough tissue for a full work up.
Answer 1: The molecular analysis performed and the amount of DNA/RNA deemed sufficient for the analysis have been specified, as suggested by the reviewer, at the end of the methods section with the following sentence. “Molecular profiling included testing for PD-L1 and for the genes reimbursed at the time of biopsy by the Italian Healthcare System (i.e., EGFR, ALK, ROS1, KRAS, BRAF). A minimum of 30 ng/l of DNA and 15 ng/l of RNA was deemed sufficient to carry out the above analysis”.
Reviewer 2 Report
This article investigated the role of an ultrasound-guided needle aspiration/biopsy (US-NAB) in the diagnosis of superficial metastasis of lung cancer, combined with/without rapid on-site evaluation (ROSE). Moreover, the author claimed that the US-NAB was safe and performed a well diagnostic yield for both tissue diagnosis and molecular profiling, which was not affected by the existence of ROSE. US-NAB could give a method for diagnosis through a safe, vessel-free path to the lesion and reduce the risk of pneumothorax. This manuscript is undoubtedly attractive but some concerns need to be addressed carefully. Below are some specific comments:
1. In the Results, it is confusing that there were 136 patients enrolled in this study but the number of sampled lesions was 144. So we’d like to know the related standard or regulation for obtaining the lesions from patients. Besides, Figure 2 was cited in the sentence “Of the 144 sampled lesions (Figure 2), superficial lymph nodes (103, 127 71.5%)…”. The content of Figure 2 seems irrelevant to the description in the results section. Considering that, the author may give more explanation and report based on the above mentioned. And the results displayed in Figure 3 are needed to be put in the part of Results at first, not Discussion.
2. In this manuscript, the authors aimed to discuss the diagnosis ability of US-NAB with/without ROSE relying on a randomized trial. However, the last paragraph in the Introduction showed that this article was to assess the impact of ROSE on diagnostic accuracy and safety outcomes in consecutive patients. These two kinds of aims were hard for us to fuse. This contradiction needed further explanation from the authors. In addition, based on the topic of investigating the diagnosis ability of US-NAB with/without ROSE, most of the Discussion was focused on the biosafety in using US-NAB instead of the diagnosis ability of US-NAB. And the safety profiles of US-NAB had mere evidence and experiments provided in this article to be verified, which was also little related to the theme of the diagnosis ability of US-NAB with/without ROSE. The author should provide more discussions to explain your results from the trial and prove your viewpoints, or provide more safety parameters to supply your results, making your discussion more convincing.
Author Response
Comment 1: In the Results, it is confusing that there were 136 patients enrolled in this study but the number of sampled lesions was 144. So we’d like to know the related standard or regulation for obtaining the lesions from patients. Besides, Figure 2 was cited in the sentence “Of the 144 sampled lesions (Figure 2), superficial lymph nodes (103, 127 71.5%)…”. The content of Figure 2 seems irrelevant to the description in the results section. Considering that, the author may give more explanation and report based on the above mentioned. And the results displayed in Figure 3 are needed to be put in the part of Results at first, not Discussion.
Answer 1: As the reviewer correctly points out, 144 lesions were sampled in 136 patients because in some of them 2 different superficial lesions were sampled in spite of one. Indeed, we had not stated in the methods section which policy was used to decide if one or more lesions were to sample in patients with multiple lesions. We have now added in the methods section the following sentence to clarify our policy: “In patients with more than one superficial lesion, the decision to sample one or more lesions was left at the discretion of the operator, who could decide based on the ROSE findings (ROSE arm) or on the gross examination of the biopsy material (US-NAB arm)”.
We totally agree with the reviewer that the reference in the text to Figures 2 and 3 was unclear and mistaken. The aim of those figures is to demonstrate that several different anatomical areas were biopsied. We put the reference to Figures 2 and 3 only in the Results section within the following new sentence: “One hundred forty-four lesions belonging to several superficial anatomical areas were sampled (Figure 2 and 3), but superficial lymph nodes (103, 71.5%), and bone metastases (18, 12.5%) were the most frequent targets”.
Comment 2: In this manuscript, the authors aimed to discuss the diagnosis ability of US-NAB with/without ROSE relying on a randomized trial. However, the last paragraph in the Introduction showed that this article was to assess the impact of ROSE on diagnostic accuracy and safety outcomes in consecutive patients. These two kinds of aims were hard for us to fuse. This contradiction needed further explanation from the authors. In addition, based on the topic of investigating the diagnosis ability of US-NAB with/without ROSE, most of the Discussion was focused on the biosafety in using US-NAB instead of the diagnosis ability of US-NAB. And the safety profiles of US-NAB had mere evidence and experiments provided in this article to be verified, which was also little related to the theme of the diagnosis ability of US-NAB with/without ROSE. The author should provide more discussions to explain your results from the trial and prove your viewpoints, or provide more safety parameters to supply your results, making your discussion more convincing
Answer 2: The reason why, in the last sentence of the introduction, we state that the aims of the studies were to assess the impact of ROSE on diagnostic accuracy and safety outcomes in consecutive patients is related to the study outcomes that we chose and that pertain to those domains: diagnostic yield for a tissue diagnosis (primary); diagnostic yield for genotyping and PD-L1 testing (secondary); and complication rate (secondary). However, as correctly underlined by the reviewer, safety outcomes may refer to multiple outcomes and may be misleading, whereas we only looked at the complication rate as outcome. As a consequence, the term safety outcomes as been replaced with “complications” in the last sentence of the introduction. In the Discussion we referred to the safety of the procedure (US-NAB from superficial lesions) because indeed we had a single complication (syncope) in 136 patients (0.73% overall complication rate).
As for the diagnostic yield for tissue diagnosis and molecular profiling, we outlined the excellent diagnostic results in extenso in the Results section and in Table 2. We also opened the discussion by stating that US-NAB of superficial metastases from lung cancer has a high diagnostic success regardless of the availability of ROSE. To avoid to be too repetitive and to magnify our good results, in the rest of the Discussion section we tried to discuss also the organizational impact of US-NAB, the feasibility of US-NAB from bone lesions, and the advantage of using large-bore needles to optimize yield.